# The Underlying Molecular Mechanisms of the Placenta Accreta Spectrum: A Narrative Review

**DOI:** 10.3390/ijms25179722

**Published:** 2024-09-08

**Authors:** Erik Lizárraga-Verdugo, Saúl Armando Beltrán-Ontiveros, Erick Paul Gutiérrez-Grijalva, Marisol Montoya-Moreno, Perla Y. Gutiérrez-Arzapalo, Mariana Avendaño-Félix, Karla Paola Gutiérrez-Castro, Daniel E. Cuén-Lazcano, Paul González-Quintero, Carlos Ernesto Mora-Palazuelos

**Affiliations:** 1Research Unit, Center for Research and Teaching in Health Sciences, Autonomous University of Sinaloa, Culiacan 80030, Mexico; eriklizarraga@uas.edu.mx (E.L.-V.); saul.beltran@uas.edu.mx (S.A.B.-O.); marisol.montoya@uas.edu.mx (M.M.-M.); perla.gutierrez@uas.edu.mx (P.Y.G.-A.); karla.gutierrez@uas.edu.mx (K.P.G.-C.); daniel.cuen@uas.edu.mx (D.E.C.-L.); 2Cátedras CONAHCYT-Food and Development Research Center (CIAD) A.C., Culiacan 80110, Mexico; erickpaulggrijalva@gmail.com; 3Faculty of Dentistry, Autonomous University of Sinaloa, Culiacan 80010, Mexico; marianaavendano@uas.edu.mx; 4Gynecology and Obstetrics Service, Women’s Hospital of Culiacan, Health Secretary, Culiacan 80020, Mexico; drpol@hotmail.es

**Keywords:** placenta accreta spectrum, trophoblast invasion, placenta acreta, placenta increta, placenta percreta

## Abstract

Placenta accreta spectrum (PAS) disorders are characterized by abnormal trophoblastic invasion into the myometrium, leading to significant maternal health risks. PAS includes placenta accreta (invasion < 50% of the myometrium), increta (invasion > 50%), and percreta (invasion through the entire myometrium). The condition is most associated with previous cesarean deliveries and increases in chance with the number of prior cesarians. The increasing global cesarean rates heighten the importance of early PAS diagnosis and management. This review explores genetic expression and key regulatory processes, such as apoptosis, cell proliferation, invasion, and inflammation, focusing on signaling pathways, genetic expression, biomarkers, and non-coding RNAs involved in trophoblastic invasion. It compiles the recent scientific literature (2014–2024) from the Scopus, PubMed, Google Scholar, and Web of Science databases. Identifying new biomarkers like AFP, sFlt-1, β-hCG, PlGF, and PAPP-A aids in early detection and management. Understanding genetic expression and non-coding RNAs is crucial for unraveling PAS complexities. In addition, aberrant signaling pathways like Notch, PI3K/Akt, STAT3, and TGF-β offer potential therapeutic targets to modulate trophoblastic invasion. This review underscores the need for interdisciplinary care, early diagnosis, and ongoing research into PAS biomarkers and molecular mechanisms to improve prognosis and quality of life for affected women.

## 1. Introduction

The placenta accreta spectrum (PAS) corresponds to a range of conditions characterized by abnormal trophoblastic invasion into the myometrium, which carries significant risks to maternal health [1]. The PAS is categorized based on the extent of invasion into the uterine wall. Placenta accreta is characterized by an invasion of less than 50% of the myometrium, increta by more than 50% invasion, and percreta by invasion through the entire myometrium [2]. The PAS is considered a high-risk condition with serious associated morbidities; therefore, the American College of Obstetricians and Gynecologists (ACOG) and the Society for Maternal-Fetal Medicine recommend these patients receive level III (subspecialty) or higher care with consistent access to interdisciplinary staff with expertise in critical care [3].

The most prevalent risk factor for PAS is a previous cesarean delivery, with the incidence of PAS increasing with the number of prior cesarean deliveries [3,4]. According to a systematic review, the rate of the PAS increases from 0.3% in women with one previous cesarean delivery to 6.74% in women with five or more cesarean deliveries [5]. Additional risk factors include advanced maternal age, multiparity, prior uterine surgeries or curettage, the manual delivery of the placenta, Asherman syndrome, postpartum endometritis, hysteroscopic surgery, endometrial ablation, and uterine artery embolization, which have all been associated with PAS disorders in subsequent pregnancies [3,6,7]. 

In this regard, cesarean delivery rates have risen substantially globally, increasing from less than 7% in the 1990s to exceeding the World Health Organization’s (WHO) recommended upper limit of 10–15% at the population level in the past two decades [8,9]. Therefore, early diagnosis is essential for management and a favorable outcome for the binomial.

This review aims to describe the molecular mechanisms driving PAS by exploring the pathological signaling pathways implicated in this disease. We will delve into the roles of critical proteins, chemokines, and other biomarkers and the role of non-coding RNAs in trophoblastic invasion. Additionally, we will examine the contributions of epigenetic modifications and genes in the involvement of key regulatory processes, including apoptosis, cell proliferation, invasion, and inflammation.

## 2. Methodology

To identify relevant information on the PAS, this review was compiled based on recent scientific literature (2014–2024) from the Scopus, PubMed, Google Scholar, and Web of Science databases. The keywords used for the literature research were “Placenta Acreta Spectrum; acreta; increta; percreta; trophoblast invasion; miRNAs, lncRNAs; genetic expression; biomarkers; signaling pathways“. We included manuscripts concerning PAS, specifically placenta accreta, increta, and percreta. We also considered research articles focusing on the most relevant cellular processes involved in PAS pathogenesis, such as angiogenesis, apoptosis, invasion, and the migration of trophoblastic cells. Publications concerning other obstetric complications, such as preeclampsia, eclampsia, and HELLP syndrome, among others, were excluded. We only included the literature in the English language. Using these criteria, this review is composed of 80 papers.

## 3. The PAS Classification and Physiopathological Features

The International Federation of Gynecology and Obstetrics (FIGOs) Committee first recognized the PAS classification for the Ethical Aspects of Human Reproduction and Women’s Health in 2011. The clinical distinction of morbid adherence of the placenta is associated with comprehensive prenatal diagnosis, a clinical approach, the correlation of treatment methods, and the frequency of severe obstetric complications [10]. Later, several retrospective and prospective studies were explicitly conducted to determine the classification influence on the results of the PAS treatment, which showed its significant impact on the choice of management [11].

In this regard, the classification of the PAS is based on the depth of placental invasion within the uterine wall and its extension of involvement. The Society for Maternal-Fetal Medicine (SMFM) developed a widely accepted classification system [12], dividing PAS into three categories: placenta accreta (PA), placenta increta (PI), and placenta percreta (PP), the features of which are presented in Figure 1.

The physiopathological features of the PAS involve a complex interplay of factors, including abnormal placentation, uterine scarring, and impaired decidualization. In normal pregnancies, the placenta separates easily from the uterine wall during childbirth due to the formation of a specialized layer known as decidua. Angiogenesis is a necessary cellular process for correct endometrial and embryonic growth and placentation. Invasion is pivotal for blastocyst differentiation into villous and extravillous trophoblasts (EVTs), acquiring pathological characteristics once they invade the decidua and myometrium [13]. Once implanted by migration, maternal uterine artery and vascular smooth muscle cells are replaced by trophoblasts through apoptosis [14]. 

However, in the PAS, the formation of the decidua is disrupted, leading to the abnormal adherence of the placenta [15]. The developmental behavior of severity in this spectrum comprises complex mechanisms, highlighting abnormal placentation, which allows trophoblast invasion to myometrium involving cellular differentiation, proliferation, and invasion in conjunction with growth factors and receptors [16]. 

Decidual deficiency is another factor that promotes placenta accreta development, and it has been linked with calcitonin and MAPK for trophoblast penetration into the endometrial epithelium [17]. Concerning trophoblastic invasion, molecules such as MMP-2 and 9, oxygen, and integrins are mainly involved [18]. In addition, it has previously been identified that PAS-associated proteins, including the vascular endothelial growth factor (VEGF), placenta growth factor (PlGF), along their respective receptors (VEGFR), epidermal growth factor receptor, c-erbB-2 oncoprotein, angiopoietin-1, angiopoietin-2, and Tie receptors [19,20,21]. Herein, cellular and molecular processes involved in the PAS are discussed.

## 4. Biomarkers Associated with Placenta Accreta Development

Clinical suspicion is based on the risk factors for the PAS and imaging findings [22]. However, the PAS is sometimes diagnosed only at delivery or by pathology since placental findings may not be visualized until the delivery time [23,24]. There is interest in identifying potential biomarkers for the PAS, particularly those that may be clinically useful and non-invasive. 

Some researchers suggest combining serum biomarker values with imaging and clinical data to improve the diagnostic performance of ultrasonography and MRI-based methods for PAS disorders. The medical literature has shown that specific peripheral blood biomarkers, including those related to angiogenesis, the immune system, beta-human chorionic gonadotropin (β-hCG), and placental-derived cell-free DNA, may increase in patients with placental invasion disorders compared to normal pregnancies [25,26]. The most relevant biomarkers for the spectrum diagnosis are presented in Table 1.

## 5. Molecular Mechanisms Involved in the Placenta Acreta Spectrum

The PAS comprehends several mechanisms involving multifactorial processes, highlighting proliferation and invasion into local tissues, similar to a tumor. Other characteristics involved in PAS physiopathology include angiogenesis induction and cell death resistance, including the epithelial-to-mesenchymal transition (EMT) [40,41,42]. Despite several reports of placental pathologies, the precise molecular mechanisms of the PAS are still poorly understood. Here, we enlist the most relevant molecular mechanisms involved in the spectrum to date (Figure 2).

### 5.1. Gene Expression

In the process of invasion, proliferation, and migration, excessive trophoblast invasion and decidual deficiency are the main pathophysiological mechanisms of PA; these processes, in combination with other mechanisms such as endometrial invasion, migration through the myometrium, among others, trigger the spectrum [43].

In this regard, the chemokine CXCL12 and its receptors, CXCR4 and CXCR7, are known to play pivotal roles in the invasion process of trophoblast cells. Consequently, the mechanisms underlying the excessive invasion of trophoblasts in patients diagnosed with PAS present an upregulation of CXCL12 and CXCR4/CXCR7 in extravillous trophoblastic cells in a dose-dependent manner, increasing their proliferative capabilities at higher expressions, and contributing to the invasion into the uterine wall. Moreover, it was noted that the regulation of trophoblast migration and invasion are related to CXCL12 and CXCR4/CXCR7 expressions in the same manner, suggesting the participation of these chemokines in the PAS development [44]. On the other hand, a study conducted by Arakaza et al. indicated that the expression of insulin-like growth factor 1 (IGF-1), fibroblast growth factor 2 (bFGF), and PlGF are important for PAS development since their expression was found to be higher in PAS placental tissue in comparison to normal placental samples [19]. Interestingly, IGF-1 expression increased among them in relation to disease severity. This phenomenon might be associated with the function of IGF-1 in trophoblastic invasion because it acts as an angiogenic growth factor since it has been demonstrated to promote tubular formation through the activation of PI3 and _MAPK pathways in human endothelial cells. In addition, IGF-1 is overexpressed in women with endometriosis, favoring invasiveness, proliferation, decreased apoptosis, and angiogenesis on ectopic cells in conjunction with HGF and IGF-1 [45,46,47]. This last point might explain the behavior of IGF-1 expression in relation to severity since it is relative to invasion rates.

The role of β-catenin has been studied regarding placental disparities; it plays an important role in the maintenance of cellular homeostasis as well as intercellular connections, which consolidate cell adhesion [48]. Concerning placental affections, in women with placenta accreta, β-catenin expression is downregulated, leading to excessive trophoblastic invasion by losing interstitial connections [49]. In addition, EMT in trophoblasts has been linked with high BAP1 expression; this mechanism was elucidated by observing *Bap1*-null mouse trophoblast stem cells (mTSCs) using CRISPR/Cas9, resulting in augmented ETM rates, which, in turn, promotes differentiation, invasion, and proliferation capacities [50].

Inflammation is another mechanism involved in the PAS since an interaction exists between the trophoblast and the uterine tissue, resulting in the exacerbated release of proinflammatory mediators [51]. In this regard, in a recent study conducted by Abdel-Hamid, Mesbah, Soliman and Firgany [15], the expression of tumor necrosis factor-alpha (TNF-α), interleukin-1 beta (IL-1β), and IL6 were overexpressed in placentas of patients with placenta accreta; interestingly, their expression levels were associated with the number of EVT, which also were higher in comparison to normal placentation tissue samples. It was argued that it might be mainly comprised TNF-α due to its ability to inhibit trophoblastic invasion. In addition, the increased co-expression of TNF-α, IL-1β, and IL6 suggest their pivotal role as pro-inflammatory mediators in placenta accreta pathogenesis.

### 5.2. The Roles of Non-Coding RNAs in the PAS

Non-coding RNAs are pivotal in regulating biological processes, including pathologic development. ncRNAs comprise regulatory molecules, of which micro RNAs (miRNAs) and long non-coding RNAs (lncRNAs) are mainly addressed in the context of biological regulation [52]. Concerning placenta-related illnesses, there have been several ncRNAs identified for their roles in these pathologic disorders, mainly in intrauterine growth retardation, preeclampsia, and PAS stages [53]. Several studies have identified valuable information about how ncRNAs present differential expression levels and their roles in regulating the PAS [54]. Nonetheless, physiopathological regulation coffered by both miRNAs and lncRNAs is still limited. In this regard, we summarize ncRNA’s most relevant pathological roles based on PAS patients.

#### 5.2.1. microRNAs

One of the key functions of miRNAs is regulating genes by mediating the degradation of mRNAs. miRNAs influence transcription and translation through two primary mechanisms: the canonical pathway, as briefly described earlier, involves the degradation of mRNAs based on the miRNA seed sequence [55]. Furthermore, circulating microRNAs have been demonstrated to play crucial roles in research and clinical settings, particularly in disease monitoring. Changes in circulating miRNAs have been linked to pathological processes, including chronic diseases, cancer, and the PAS [56,57].

miRNA behavior in PAS development is still poorly explored. Nonetheless, there is evidence that its role is related to cellular processes that accompany pathological development. In this regard, miR-7-5p plays an important role in trophoblast invasion since its overexpression converged in a significant diminution of cell invasion in HTR-8/SVneo cells; moreover, its downregulation results in an increase in SNAIL, SLUG, TWIST, and vimentin expression, promoting EMT and trophoblast invasion [58]. In extravillous trophoblast cells, miR-519d is highly expressed, and its main activity is to control migration by suppressing CXCL6, FOXL2, and NR4A2; in addition, MMP2 is a target gene of miR-519d, which suggests that it is involved in trophoblast invasiveness [59]. Murrieta-Coxca et al. [60] identified a plethora of deregulated expressed miRNAs in placenta accreta tissue; miR-24-3p, miR-193b-3p, miR-331-3p, miR-376c-3p, miR-382-3p, miR-495-3p, miR-519d-3p, and miR-3074-5p were overexpressed in PAS tissue while miR-106b-3p, miR-222-3p, miR-370-3p, miR-454-5p, and miR-3615-3p presented a downregulation in PAS tissue. Through biological pathway analysis, a significant reduction in NF-kB mRNA was identified, which was also confirmed in PAS samples, suggesting their influence of higher invasive capacities through the actions of miR-382-3p and miR-495-3p. 

On the other hand, miR-106b-3p, miR-222-3p, and miR-519d-3p target PTEN, which controls the cell cycle, mainly in trophoblast proliferation and migration. The microRNA 1296-5p is overexpressed in the tissue of the PAS patient; its role is presumed to provide the regulation of apoptosis since its overexpression affected AGGF1, which, in turn, inhibits P53 and Bax expression while also increasing the expression of Bcl-2 protein [61]. In placenta accreta tissues, the overexpression of miR-518b was positively associated with OPN and VEGF, playing essential roles in regulating villous trophoblast cell migration, invasion, and adhesion [62]. In placenta accreta tissues, MCL1 expression is higher than in normal tissue, specifically in intermediate trophoblast cells, which inhibits apoptosis. The gene MCL1 has been reported as a target gene for both miR-29a/b/c and miR-125a; its cellular role results in the promotion of the apoptosis of trophoblast cells while downregulating MCL1 expression [63,64].

#### 5.2.2. Long Non-Coding RNAs

Long non-coding RNAs (lncRNAs) are crucial because they play a vital role in maintaining homeostasis in biological processes. However, lncRNAs also contribute significantly to the development of diseases. Their biogenesis involves a comprehensive genomic system, including promoters, enhancers, and intergenic regions in eukaryotic genomes [65].

The roles exerted by lncRNAs are also pivotal in regulating cellular processes concerning placental invasive behaviors. In this regard, H19 downregulation can sponge miRNA let-7; as a consequence, TβR3 (type III TGF-β receptor) expression is negatively affected, promoting the invasion of EVT by increasing migration and invasion rates [66]. The high expression of lncRNA SNHG6 enhances the invasion of the human extravillous trophoblast HTR-8/SVneo cell line; this mechanism is argued to occur due to the SNHG6/miR-101-3p/OTUD3 regulatory axis [67]. Another regulatory axis in the PAS has been linked to lncRNA SNHG16 since its expression was diminished in placental affections, like preeclampsia. Nonetheless, its behavior has been evaluated in HTR-8/SVneo cells and is often overexpressed, resulting in the creation of cell proliferation, migration, and invasion as well as the inhibition of apoptosis, thus sponging miR-218-5p which, in turn, suppresses LASP1, which is a protein that facilitates cell invasion in diverse types of malignancies [68].

In addition, lncRNA uc.187 aberrant expression has been linked with higher proliferation rates, invasion, and lower apoptotic activities in HTR-8/SVneo cells. Regarding these mechanisms, they are activated by the increasing expression of MMP-2/-9 and PCNA/Ki-67 proteins, leading to enhanced invasion and proliferation, respectively; on the contrary, uc.187 overexpression affects the Bcl-2 protein, converging in a reduction in cell death in trophoblast cells [69].

Several reports indicate the cellular behavior of ncRNAs in placental pathologies. Since placenta accreta is an emerging pathology, and many aspects are still unknown. Comprehending molecular mechanisms can aid in developing early diagnostics and treatment in affected women.

## 6. Aberrant Signaling Pathways in the PAS

Several signaling pathways have been involved in the etiology of the PAS, such as the Notch signaling pathway, which is essential in regulating angiogenesis through the overexpression of periostin (POSTN) in the HUVEC cell line during the neovascularization process in conjunction with HES1 and Hey1 overexpression [70,71]. The above-mentioned pathway has been linked to an exacerbated hemorrhage in the PAS, mainly attributed to hypervascularity in the uteroplacental and utero–bladder interfaces [72].

Interestingly, AGGF1 is downregulated in PAS samples, and its deletion in human trophoblast HTR8/SVneo cells enhances mechanisms related to the invasive phenotype of PAS, such as proliferation, invasion, and migration. It also represses apoptosis by downregulating P53 and Bax and stimulating Bcl-2 overexpression, highlighting AGGF1 as a regulator of the P53 signaling axis [61].

Furthermore, the invasive trait of trophoblastic cells has been related to YKL-40 enhancer activity since it is overexpressed both in PAS samples and in vitro using HTR8/SVneo cells, promoting proliferation, migration, and invasion, but also inhibiting apoptosis through the activation of the Akt/MMP9 signaling pathway [73]. Additionally, STAT3, p38, and JNK pathways have been related to trophoblast invasion in the PAS since FYN stimulates the activation of STAT3, p38, and JNK through phosphorylation [74]. In addition, LAMC2 is overexpressed in placental cells, and in vitro has shown an increase in cell proliferation, invasion, and migration but inhibited apoptosis, accompanied by the elevated protein expression of MMP2, MMP9, and phosphorylated Akt (pAkt), which means that LAMC2 is implicated in the pathogenesis of PAS by activating the PI3K/Akt/MMP2/9 signaling pathway to stimulate trophoblast over-invasion [75].

Duan et al. [76] found that immunoblotting and qPCR analysis in abnormally invasive placentas (AIP), including PA, PI, or PP, showed that CCN3 overexpression is accompanied by high levels of p53, p16, p21, cyclin D1, Notch-1 cleaved, pFAK, pAkt, and pmTOR, as well as low levels of pRb, suggesting that CCN3 mediates senescence by cell cycle arrest through the activation of the FAK-Akt-mTOR pathway and cleaved Notch-1/p21, contributing to increasing the invasion properties of EVT.

Furthermore, growth factor signaling pathways, such as the macrophage-induced netrin-1/DCC/VEGF signaling pathway, have been implicated in trophoblastic angiogenesis in the PAS tissues through netrin-1, DCC receptors, VEGF overexpression, and the high recruitment of macrophages compared to normal placental tissue [77]. Likewise, using in vitro assays using the gestational choriocarcinoma cell line JEG-3 and the trophoblast cell line HTR-8/SVneo, it was found that the non-canonical TGF-β-UCHL5-Smad2 signaling pathway is essential for the invasion of EVTs, a critical step in placental development, in which Smad1/5/9 are the governing factors. In addition, the TGF-β-UCHL5-Smad2 pathway is also regulated by the ERK signaling pathway since it promotes angiogenesis and vascularization, and the alteration of these signaling pathways can cause abnormal placental invasion and angiogenesis, which leads to triggering PA [78]. Another novel mechanism described as involved in the development of the PAS consists of suppressing Wnt-β-catenin/VEGF signaling through the pigment epithelium-derived factor (PEDF), which is downregulated in PAS tissues [79]. In addition, it has been reported that PEDF overexpression inhibits EVTs proliferation, invasion, and angiogenesis and induces ferroptosis, a newly described form of regulated cell death creating a favorable scenario for adequate trophoblastic invasion; -

## 7. Perspectives and Conclusions

We provide an overview of available therapeutic strategies that could aid in the early diagnosis of the PAS. It is essential to comprehend the molecular mechanisms underlying the PAS, particularly the signaling pathways involved in the early stages accompanying this pathology, which is addressed by trophoblastic invasiveness and angiogenesis. Researchers aim to develop specific biomarkers that can effectively prevent the PAS. Moreover, exploring the signaling pathways involved could provide valuable tools to develop targeted therapies that treat the PAS, which is mainly attributed to TGF-β-UCHL5-Smad2, ERK, and Wnt-β-catenin/VEGF pathways, which play significant roles in the development and progression of the PAS. Therapeutic strategies targeting these pathways could potentially inhibit abnormal placental invasion and angiogenesis. 

Additionally, implementing other molecules as novel biomarkers, such as PEDF, is pivotal to reducing the severity of the PAS since it is downregulated in pathological tissues. Otherwise, the overexpression of PEDF in vitro has been shown to inhibit the proliferation, invasion, and angiogenesis of EVTs and induce ferroptosis, suggesting a promising therapeutic strategy.

Physiopathology accompanying the spectrum is a multifactorial interplay of abnormal placentation, uterine scarring, and impaired decidualization, resulting in abnormal placental adherence [80]. Promising advances in identifying new biomarkers, such as AFP, sFlt-1, β-hCG, PlGF, and PAPP-A, could aid in the early detection and correct management of affected women [21,27,37]. In addition, the comprehension of molecular mechanisms, such as genetic expression and the roles of non-coding RNAs, are pivotal for unveiling the complexities of the PAS [43,53,54].

In addition, identifying aberrant signaling pathways, such as Notch, PI3K/Akt, STAT3, and TGF-β, offers potential targets for therapeutic interventions to modulate trophoblastic invasion and improve patient outcomes since they play pivotal roles in the invasiveness of trophoblastic cells contributing to the PAS pathogenesis [71,74]

Overall, this review highlights the importance of interdisciplinary care, early diagnosis, and a comprehensive understanding of the molecular underpinnings of PAS. Continued research into PAS’s biomarkers and molecular mechanisms is crucial for developing effective diagnostic and therapeutic strategies, ultimately improving affected women’s prognosis and quality of life.

## Figures and Tables

**Figure 1 ijms-25-09722-f001:**
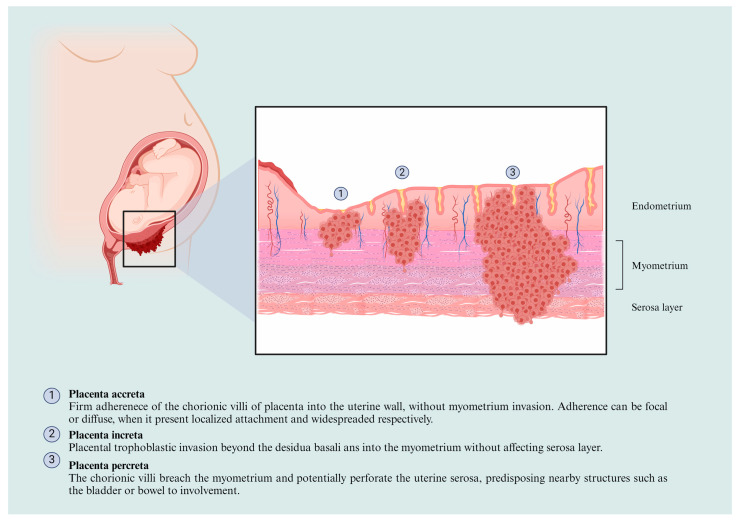
The PAS classification. The invasive cells start at the uterine wall (1); the severity of the spectrum is relative to the invasive depth, affecting the myometrium layer (2) and subsequently invading the serosa layer (3). Figure created by https://www.biorender.com/. Accessed on 7 May 2024.

**Figure 2 ijms-25-09722-f002:**
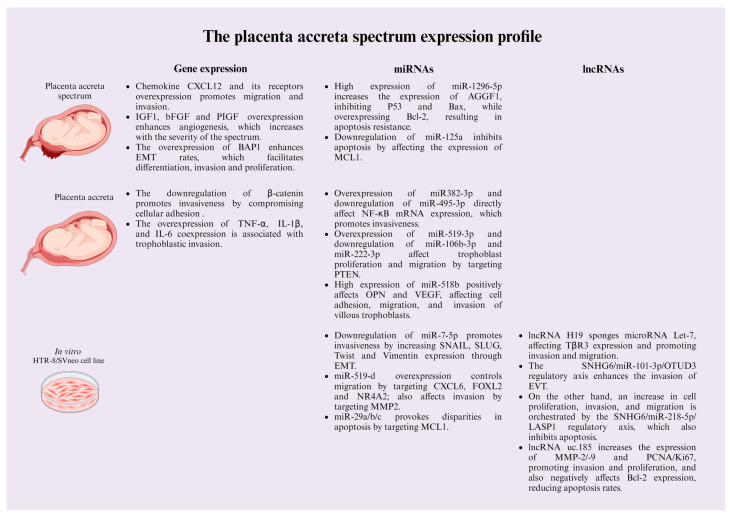
The PAS expression profile. The placenta accreta spectrum is accompanied by changes in the expression of several genes and non-coding RNAs, which are crucial in the dysregulation of cellular processes, such as migration, invasiveness, proliferation, and resistance to cell death, which facilitates trophoblastic invasion into the uterine layers. Figure created by https://www.biorender.com/. Accessed on 7 May 2024.

**Table 1 ijms-25-09722-t001:** Biomarkers related to the PAS.

Biomarker	Source	Findings	References
Alpha-fetoprotein (AFP)	Maternal serum	APF showed sensitivity and specificity of 71 and 46%, respectively, to serve as a biomarker for pathological placentation, specifically in women with placenta previa and acreta in the second trimester. Thus, a high level of AFP can be used as a cause for suspicion in high-risk pathological placentation.	[27]
		Maternal serum AFP levels were associated with PAS patients; it was established as a predictor for PAS patients that require hysterectomy with 85.94% sensitivity and 71.43% specificity.	[28]
Soluble fms-like tyrosine kinase-1 (sFlt-1)	Maternal serum	Third trimester sFlt-1 serum levels were decreased in PAS-affected women, respectively, with pathological severity.	[29]
	Maternal plasma	Concentrations of sFlt-1 were lower in patients with PAS than those with normal placentation, with 90.0% sensitivity and 82.0% specificity. The lower concentrations were also associated with intraoperative blood loss.	[21]
β human chorionic gonadotrophin (β-hCG)	Maternal plasma or serum	The elevated concentration of β-HCG in serum may be appropriate for the prenatal diagnosis of placenta accreta, which suggests the relationship between the risk of PAS and the first trimester.	[30]
	Maternal serum	hCG showed a sensitivity and specificity of 53 and 68%, respectively, to serve as a biomarker for pathological placentation. Higher levels of hCG can be used as a cause for suspicion in high-risk pathological placentation.	[27]
	Maternal plasma cell-free β-hCG mRNA	Cell-free β-hCG mRNA concentrations were significantly elevated in women with placenta accreta. This suggests that β-hCG mRNA levels might be a marker for identifying women with placenta accreta likely to require hysterectomy.	[31]
Placental growth factor (PlGF)	Maternal plasma	Concentrations of PlGF were higher in patients with PAS than those with normal placentation, with 86.0% sensitivity and 93.0% specificity. Higher concentrations were also associated with intraoperative bleeding.	[21]
	Maternal serum	PIGF serum levels were higher in PAS severity groups than in normal placentation patients, including placenta previa patients, suggesting these levels are a predictor criterion exclusive for PAS patients with 83% sensitivity and 82% specificity.	[32]
	Maternal serum and placental bed tissues	High serum levels and high placental bed expression in placenta previa patients with PAS disorders were explored. PlGF serum levels might predict PAS affection, excepting the severity grade based on FIGO.	[19]
Pregnancy-associated plasma protein-A (PAPP-A)	Maternal serum	Increased first-trimester serum was positively associated with placenta accreta, suggesting the potential role of PAPP-A as a biomarker in identifying pregnancies at high risk for placenta accreta.	[30,33,34,35,36]
		A significant correlation was found between PAPP-A levels and blood loss volume. This suggests that first-trimester PAPP-A levels may be useful for the early prediction of pathological blood loss at delivery in pregnant women with PAS and for recognizing a high-risk group for PAS.	[37]
Human placental lactogen mRNA (hPL mRNA)	Maternal plasma	The expression of hPL mRNA is elevated in the plasma of women diagnosed with placenta previa and invasive placenta between 28 and 32 weeks of gestation.	[38]
		The multiple of the median (MoM) for hPL mRNA was significantly higher in the placenta accreta group compared to the control and placenta previa groups.	[39]

## Data Availability

No new data were created or analyzed in this study. Data sharing is not applicable to this article.

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
