# Peer review of "The Underlying Molecular Mechanisms of the Placenta Accreta Spectrum: A Narrative Review"

_ijms, 2024, doi:10.3390/ijms25179722_

Round 1

Reviewer 1 Report

Comments and Suggestions for Authors

The manuscript offers an overview of some of the molecular mechanisms involved in placenta accreta spectrum (PAS) disorders. It covers biomarkers, gene expression, non-coding RNAs, and signaling pathways, demonstrating a thorough understanding of the subject matter. The authors have compiled information from recent scientific literature (2014-2024), ensuring the relevance and timeliness of the content.

However, there are areas where the manuscript must be significantly improved to have a publishable manuscript. Overall, English proficiency and grammar must be substantially improved, as significant problems are found throughout the text, as detailed above. The authors need to perform a comprehensive editing process in several sections of the manuscript, starting with the abstract. From lines 28 and below, the structure of the abstract is confusing, i.e., some concepts in lines 28 and 29 are already described above. In line 23, the redaction should be revised in: “ it explored…”

 The introduction is well written, but some parts need improvement; for example, section 3 could be before the methodology section, as it describes some general aspects of the pathology.

Line 27: remove “more recently”, there is no information in the references about the timeline of the risk factors description

Figure 1:  Adding tags to the different structures of the endometrium/placenta would improve the diagram

Lines 103-111: This is a key paragraph for the review, a more detailed description and explanation of the involved biological processes should be included in order to understand the following review.

The methodology section could be more detailed, specifying the search terms used, inclusion and exclusion criteria, and the process of selecting relevant articles. This would enhance the transparency and reproducibility of the review.

My main concern starts from section 4, the quality of writing is too heterogeneous and lacks rigor.

Line 114-114: that is already explained above

Line 115: Is the “Clinical suspicion”  a proper term? Won’t be imaging findings confirmatory?

Lines 133-139: Again, repeated ideas in the text

Line 150: Please explain “… cells in a dose-dependent manner”; the statement is incomplete, and more detail is needed to understand the idea.

Line 151: migration is repeated

Lines 158-159: a more detailed explanation of the link between trophoblastic invasion and angiogenesis is required

Line 164: Please use a uniform term to refer to the PAS, informal terms like “accreta”  “creta “ (lines 175, 179, 226,226) should be avoided

Lines 164-166: why is downregulation of β-catenin an important cause? Are there other causes more or less important? I think that the term important is not adequate

Line 169: “.. which in turn regulates differentiation…” up-regulate or down-regulate? Please explain

 Lines 216-218: please check grammar; it’s hard to understand the idea in those lines as written in the manuscript

Line 231: delete the extra “s”

Line 238: Please use standar nomenclature for miRNA

Line 239: EVT abbreviation was introduced above in the text (line 107)

Lines 254-257: is this paragraph related to PAS? If not, it should be deleted.

Lines 264-268: Please check this paragraph, it is redacted in an unusual form and can induce to misinterpretations

Line 202: correct “PA”

Lines 306-308: please explain the significance of this phrase

Line 314-315: please check grammar, the excessive use of thus or besides is not adequate

Line 316: “… manage PAS are…” It’s hard to understand the idea as written

In general, the perspective and conclusions section is weak and needs a better design and flow of the concepts, a resume of the review is not necessary, this section could be strengthened by summarizing the key findings of the review and highlighting their clinical relevance. The authors could also discuss potential future research directions and the need for further investigation into the molecular mechanisms of PAS.

Overall, the manuscript is informative, providing a valuable resource for researchers and clinicians interested in PAS. However, addressing the areas mentioned above for improvement would enhance its clarity, rigor, and impact.

Comments on the Quality of English Language

Some sections require extensive grammar improvement

Author Response

The manuscript offers an overview of some of the molecular mechanisms involved in placenta accreta spectrum (PAS) disorders. It covers biomarkers, gene expression, non-coding RNAs, and signaling pathways, demonstrating a thorough understanding of the subject matter. The authors have compiled information from recent scientific literature (2014-2024), ensuring the relevance and timeliness of the content.

However, there are areas where the manuscript must be significantly improved to have a publishable manuscript. Overall, English proficiency and grammar must be substantially improved, as significant problems are found throughout the text, as detailed above. The authors need to perform a comprehensive editing process in several sections of the manuscript, starting with the abstract. From lines 28 and below, the structure of the abstract is confusing, i.e., some concepts in lines 28 and 29 are already described above. In line 23, the redaction should be revised in: “ it explored…”

We appreciate your comments and suggestions; they were taken into consideration, and we edited the abstract section.

 The introduction is well written, but some parts need improvement; for example, section 3 could be before the methodology section, as it describes some general aspects of the pathology.

Line 27: remove “more recently”, there is no information in the references about the timeline of the risk factors description

We omitted the term

Figure 1:  Adding tags to the different structures of the endometrium/placenta would improve the diagram

We appreciate your suggestion; we added corresponding tags

Lines 103-111: This is a key paragraph for the review, a more detailed description and explanation of the involved biological processes should be included in order to understand the following review.

This paragraph has been edited and a more detailed description has been added

The methodology section could be more detailed, specifying the search terms used, inclusion and exclusion criteria, and the process of selecting relevant articles. This would enhance the transparency and reproducibility of the review.

We appreciate your suggestions about this section, we added the recommended criteria

My main concern starts from section 4, the quality of writing is too heterogeneous and lacks rigor.

Line 114-114: that is already explained above

We omitted repeated information

Line 115: Is the “Clinical suspicion”  a proper term? Won’t be imaging findings confirmatory?

It is referred to as suspicion based on risk factors and aided by imaging findings. Nonetheless, affected pregnant women can be unnoticed due to the invasion grade.

Lines 133-139: Again, repeated ideas in the text

We removed the alleged repeated idea; we conserved the introductory paragraph with the main characteristics. In this part, we first mentioned some information like cell death resistance or EMT

Line 150: Please explain “… cells in a dose-dependent manner”; the statement is incomplete, and more detail is needed to understand the idea.

We mentioned

Line 151: migration is repeated

Repeated term was deleted

Lines 158-159: a more detailed explanation of the link between trophoblastic invasion and angiogenesis is required

More detailed information was added

Line 164: Please use a uniform term to refer to the PAS, informal terms like “accreta”  “creta “ (lines 175, 179, 226,226) should be avoided

We attended your suggestion

Lines 164-166: why is downregulation of β-catenin an important cause? Are there other causes more or less important? I think that the term important is not adequate

This suggestion was attended, and the text was edited

Line 169: “.. which in turn regulates differentiation…” up-regulate or down-regulate? Please explain

It was mentioned that these mechanisms are affected after BAP1 knock out; the term regulates was replaced by promotes

 Lines 216-218: please check grammar; it’s hard to understand the idea in those lines as written in the manuscript

This part was checked and corrected

Line 231: delete the extra “s”

Deleted

Line 238: Please use standar nomenclature for miRNA

Done

Line 239: EVT abbreviation was introduced above in the text (line 107)

The complete term was deleted

Lines 254-257: is this paragraph related to PAS? If not, it should be deleted.

It was added concerning the effect of trophoblastic invasion. Nonetheless, we opted to remove this citation since it was over preeclampsia conditions

Lines 264-268: Please check this paragraph, it is redacted in an unusual form and can induce to misinterpretations

This paragraph was edited

Line 202: correct “PA”

This term was not erroneous, it is referred to as Placenta Acreta

Lines 306-308: please explain the significance of this phrase

This phrase has been corrected

Line 314-315: please check grammar, the excessive use of thus or besides is not adequate

Grammar checked

Line 316: “… manage PAS are…” It’s hard to understand the idea as written

The term was replaced

In general, the perspective and conclusions section is weak and needs a better design and flow of the concepts, a resume of the review is not necessary, this section could be strengthened by summarizing the key findings of the review and highlighting their clinical relevance. The authors could also discuss potential future research directions and the need for further investigation into the molecular mechanisms of PAS.

Overall, the manuscript is informative, providing a valuable resource for researchers and clinicians interested in PAS. However, addressing the areas mentioned above for improvement would enhance its clarity, rigor, and impact.

Reviewer 2 Report

Comments and Suggestions for Authors

This review provides comprehensive insight on the molecular background of placenta accreta spectrum. It is very informative and interesting. However, some minor issues should be addressed before final acceptance

Line 16-18: regarding the definition of placental invasion types, it is appropriate to define these types in relation to the uterine wall not to the myometrium, as myometrium is a single layer from the uterine wall not all the uterine wall.

 Key words are not well-presentable. It should be diverse, differ from those words in the title, not repeatable. For example, just one word “placenta” instead of these words of placenta in the key words. It helps the expanding of the article in the search engines.

Line 41-43: add a relevant reference.

Line 43-46: the same as before. The authors accepted the classification of The Society for Maternal-Fetal Medicine and affirm that the placenta accreta not involved the myometrium. How the authors defined the placenta accreta as less than 50% myometrial invasion. Please recheck.

Figure 1: how does this figure designed. Please add the source or the program.

Line 114: the authors already mentioned the placenta accreta spectrum and its abbreviation early in the commencement of the introduction section. Please use the abbreviation only afterward.

Line 114-115: “The Placenta accreta spectrum (PAS) encompasses the disorders of abnormal placental adherence, including placenta accreta, increta, and percreta” the authors mentioned this information more than two times before. It is enough.

Table 1: is not well-organized and presented.

Line 132: use the abbreviation only.

Line 153: it is enough to use the first author followed by et al. not mention all these authors.

Line 263: the same

Line 265: add a relevant reference after “angiogenesis”.

Line 326-332: this paragraph is repeated, please omit.

In the conclusion section, please do not repeat what you mentioned earlier in the review.

344-348: omit this paragraph, it is useless.

The review needs more graphs to ease the data delivering rather than these word blocks.

Author Response

This review provides comprehensive insight on the molecular background of placenta accreta spectrum. It is very informative and interesting. However, some minor issues should be addressed before final acceptance

Line 16-18: regarding the definition of placental invasion types, it is appropriate to define these types in relation to the uterine wall not to the myometrium, as myometrium is a single layer from the uterine wall not all the uterine wall.

We made these changes according to your suggestions

 Key words are not well-presentable. It should be diverse, differ from those words in the title, not repeatable. For example, just one word “placenta” instead of these words of placenta in the key words. It helps the expanding of the article in the search engines.

Line 41-43: add a relevant reference.

Reference added

Line 43-46: the same as before. The authors accepted the classification of The Society for Maternal-Fetal Medicine and affirm that the placenta accreta not involved the myometrium. How the authors defined the placenta accreta as less than 50% myometrial invasion. Please recheck.

This part was adequate in following your suggestion

Figure 1: how does this figure designed. Please add the source or the program.

This part was added to the figure

Line 114: the authors already mentioned the placenta accreta spectrum and its abbreviation early in the commencement of the introduction section. Please use the abbreviation only afterward.

The term had been changed along the text

Line 114-115: “The Placenta accreta spectrum (PAS) encompasses the disorders of abnormal placental adherence, including placenta accreta, increta, and percreta” the authors mentioned this information more than two times before. It is enough

We appreciate your observations; we deleted this part

Table 1: is not well-organized and presented.   

We appreciate your suggestion. We attended and changed the table organization.

Line 132: use the abbreviation only.

This part has been attended

Line 153 (165): it is enough to use the first author followed by et al. not mention all these authors.

The reference has been corrected

Line 263: the same

Line 265: add a relevant reference after “angiogenesis”.

Reference has been added

Line 326-332: this paragraph is repeated, please omit.

The mentioned paragraph has been deleted.

In the conclusion section, please do not repeat what you mentioned earlier in the review.

344-348: omit this paragraph, it is useless.

We removed the mentioned paragraph

The review needs more graphs to ease the data delivering rather than these word blocks.

We appreciate your suggestion; we added a second figure to the manuscript

Reviewer 3 Report

Comments and Suggestions for Authors

This narrative review reviews the current literature of the last ten years to summarise the molecular mechanisms driving PAS.  Overall, the topic is relevant, and the manuscript is well structured.  However, as with any narrative review the search strategy was unclear.  Overall, there is a lot of repetition in the manuscript and then there are some specific comments:  

1.     Title: the title is misleading, and I would suggest a title similar tot “Underlying molecular mechanisms of placenta accreta spectrum, a narrative review. “

2.     Abstract

2.1.  Is too long and doesn’t state the nature of the review. Please shorten and make it clear to the point. 

3.     Methods

3.1.  Please define more clearly the methods behind this review. Is this a state if the art narrative review? If so, what is the justification for inclusion and exclusion Criteria. For instance, why only from 2014. 

4.     Perspective and conclusions:

4.1. There is a lot of repetition in this section, especially from the introduction (paragraph 3 and 4 of this section). Please keep to a coherent message. 

Author Response

This narrative review reviews the current literature of the last ten years to summarise the molecular mechanisms driving PAS.  Overall, the topic is relevant, and the manuscript is well structured.  However, as with any narrative review the search strategy was unclear.  Overall, there is a lot of repetition in the manuscript and then there are some specific comments:  

  1. Title: the title is misleading, and I would suggest a title similar to “Underlying molecular mechanisms of placenta accreta spectrum, a narrative review. “

dear reviewer, we appreciate the suggestion and made changes to review the title

  1. Abstract

2.1.  Is too long and doesn’t state the nature of the review. Please shorten and make it clear to the point. 

We adjusted the Abstract section regarding your suggestions

  1. Methods

3.1.  Please define more clearly the methods behind this review. Is this a state if the art narrative review? If so, what is the justification for inclusion and exclusion Criteria. For instance, why only from 2014. 

  1. Perspective and conclusions:

4.1. There is a lot of repetition in this section, especially from the introduction (paragraph 3 and 4 of this section). Please keep to a coherent message. 

This section was attended considering your suggestions

Round 2

Reviewer 1 Report

Comments and Suggestions for Authors

The current version of the manuscript shows  substantial improvements, my concerns have been clarified and in my opinion the manuscript must be accepted.

minor comments:

Line 164: remove the aditional "_" before MAPK

Comments on the Quality of English Language

no comments